



# Assessing the coastal hazard of medicane Ianos through ensemble modelling

Christian Ferrarin[1], Florian Pantillon[2], Silvio Davolio[3], Marco Bajo[1], Mario Marcello Miglietta[4], Elenio Avolio[5], Diego S. Carrió[6], Ioannis Pytharoulis[7], Claudio Sanchez[8], Platon Patlakas[9], Juan Jesús González-Alemán[10], and Emmanouil Flaounas[11]

[1]CNR - National Research Council of Italy, ISMAR - Marine Sciences Institute, Venice, Italy
[2]Laboratoire d'Aérologie, Université de Toulouse, CNRS, UPS, IRD, Toulouse, France
[3]CNR - National Research Council of Italy, ISAC - Institute of Atmospheric Sciences and Climate, Bologna, Italy
[4]CNR - National Research Council of Italy, ISAC - Institute of Atmospheric Sciences and Climate, Padua, Italy
[5]CNR - National Research Council of Italy, ISAC - Institute of Atmospheric Sciences and Climate, Lamezia Terme, Italy
[6]Department of Physics, Universitat de les Illes Balears, Palma, Spain
[7]Department of Meteorology and Climatology, School of Geology, Aristotle University of Thessaloniki, Thessaloniki, Greece
[8]Met Office, Exeter, UK
[9]Department of Physics, National and Kapodistrian University of Athens, Athens, Greece
[10]Spanish State Meteorological Agency, Madrid, Spain
[11]Institute of Oceanography, Hellenic Centre for Marine Research, Athens, Greece

**Correspondence:** Christian Ferrarin (c.ferrarin@ismar.cnr.it)

**Abstract.** On 18 September 2020, medicane Ianos hit the western coast of Greece resulting in flooding and severe damage at several coastal locations. In this work, we aim at evaluating its impact on sea conditions and the associated uncertainty through the use of an ensemble of numerical simulations. We applied a coupled wave-current model to an unstructured mesh representing the whole Mediterranean Sea, with a grid resolution increasing in the Ionian Sea along the cyclone path and the landfall area. To investigate the uncertainty of modelling sea levels and waves for such an intense event, we performed a multimodel ensemble of ocean simulations using several coarse (10 km) and high-resolution (2 km) meteorological forcings from different mesoscale models. The performance of the ocean and wave models was evaluated against observations retrieved from fixed monitoring stations and satellites. All model runs emphasized the occurrence of severe sea conditions along the cyclone path and at the coast. Due to the rugged and complex coastline, extreme sea levels are localised at specific coastal sites. However, numerical results show a large spread of the simulated sea conditions for both the sea level and waves highlighting the large uncertainty in simulating this kind of extreme event. The multi-model / multi-physics approach allows us to assess how the uncertainty propagates from meteorological to ocean variables and the subsequent coastal impact. The ensemble mean and standard deviation were combined to prove the hazard scenarios of the potential impact of such an extreme event to be used in a flood risk management plan.



## 1 Introduction

Destructive sea storms driven by intense cyclones represent one of the main threats to the Mediterranean coastal communities causing flooding, beach erosion and damage to infrastructures and cultural heritages (Lionello et al., 2019; Flaounas et al., 2022). Most of these cyclones are synoptic-scale events and may produce severe sea conditions and large sea level anomalies along the coast of the Mediterranean Sea (Lionello et al., 2006, 2019). The Mediterranean basin occasionally hosts also intense mesoscale vortices that evolve into tropical-like cyclones called "medicanes" (an abbreviation of Mediterranean hurricanes Emanuel, 2005; Miglietta, 2019). Although they are more intense over the sea and quickly dissipate over land, their landfall may be associated with destructive extreme events, such as heavy precipitation, windstorms, flooding and marine storminess (Patlakas et al., 2021; Flaounas et al., 2022). The associated strong winds, high waves and storm surges can directly affect coastal infrastructure and ships en route (Sánchez-Arcilla et al., 2016). Indeed, several ship accidents caused by rough waves during intense cyclones are reported in the Mediterranean Sea (Bertotti and Cavaleri, 2008; Cavaleri et al., 2012).

The accurate numerical modelling of intense weather extremes events in the Mediterranean Sea still represents a challenge. Research from the past decade has focused on extreme precipitation events in the region and showed that high grid resolution, of the order of a few km, is required to describe small-scale processes such as intense convective activity or the interaction with the local orography, while the representation of air-sea exchanges is crucial (e.g. Ducrocq et al., 2014). In contrast, there is no wide consensus concerning modelling requirements for intense Mediterranean cyclones, including medicanes, where both large-scale dry dynamics and small-scale moist processes play a key role (Miglietta, 2019; Flaounas et al., 2021).

The awareness of the prediction uncertainties and errors for these intense weather events has led many operational and research forecasting systems around the world to move toward numerical forecasts based on a probabilistic concept: the ensemble technique (Leutbecher and Palmer, 2008; Cloke and Pappenberger, 2009; Slingo and Palmer, 2011). The most common ensemble technique approaches used operationally are based on generating perturbed initial conditions by (a) using different parameterization schemes with the same model (multi-physics ensemble approach; Jankov et al., 2017) or (b) using a set of different numerical models (multi-model ensemble approach; Rozante et al., 2014). An ensemble approach is particularly useful in coastal risk forecasting systems since it not only provides a quantitative measure of the uncertainty associated with predictions, which can be particularly large for extreme storms but also may allow estimating the probabilities of different outcomes (e.g., flooding threshold; Zou et al., 2013; Ferrarin et al., 2020). Moreover, coastal flooding and damage risks of these intense cyclones are strongly associated with the geomorphological characteristics of the considered coastal segment, especially along complex coastlines with many islands and headlands (Bosom and Jiménez, 2011; De Leo et al., 2019). The simulation of such extreme sea conditions at the coast requires numerical models with sufficiently refined horizontal grids to represent complex morphological and bathymetric features, as well as several anthropogenic constructions (e.g., piers, harbours, breakwaters, jetties) present along the coast. From this point of view, using unstructured grids allows high grid resolution simulations in coastal areas, thus resolving processes at different spatial scales and maintaining computational efficiency compared to regular grids (Umgiesser et al., 2022).



One of the most destructive tropical-like cyclones, named Ianos, occurred in the central Mediterranean basin in September 2020 causing heavy rains, damaging winds and storm surge, particularly in southern Italy and Greece (Zekkos et al., 2020; Lagouvardos et al., 2022; Zimbo et al., 2022). This event was selected as the first testbed within the research activities promoted within the international COST Action CA19109 MedCyclones (https://medcyclones.utad.pt/). The main objective of this project was to evaluate the performance of different meteorological model simulations, as well as to investigate the relevant physical mechanisms responsible for such an intense storm. Here, the numerical experiments are exploited to simulate sea conditions and coastal hazards associated with this kind of extreme event. To investigate how the model uncertainty associated with the reproduction of such a severe event propagates from the atmosphere to the marine coastal areas, we performed an ensemble of ocean simulations forced by atmospheric fields from a suite of numerical weather models. The ensemble approach allows the assessment of the potential coastal hazard associated with a medicane identifying the coastal areas impacted by severe waves and storm surges. The developed methodology can be directly implemented in an early warning procedure for providing an estimation of the peak sea storm conditions to be used in coastal risk management.

This study is structured as follows. Section 2 presents the modelling framework, Section 3 gives the results of atmospheric and ocean models, Section 4 discusses potential coastal hazards, and Section 5 concludes the paper.

## 2   The modelling framework

### 2.1   Meteorological modelling

In this study, we used an ensemble of meteorological models as a forcing condition for the ocean simulations. To reproduce the meteorological characteristics of medicane Ianos, we performed a dynamic downscaling of the operational deterministic analysis of the European Centre for Medium-Range Weather Forecasts global Integrated Forecasting System (ECMWF IFS). Two different horizontal grid resolutions, 10 km (about 0.1 degrees) and 2 km (about 0.02 degree), over a domain covering the central Mediterranean region (the black box indicated in the top panel of Fig. 1) were used. The deterministic ECMWF IFS atmospheric model (cycle 47r1) has approximately 9 km horizontal resolution and 137 vertical levels, of which 20 below 1000 m (see the full documentation available at https://www.ecmwf.int/en/forecasts/about-our-forecasts/evolution-ifs-cycles/summary-cycle-47r1). The operational analyses are based on the incremental form of the 4-Dimensional Variational Data Assimilation scheme (4D-Var; Rabier et al., 2000) and are available at 6 hours intervals (i.e., at 00, 06, 12 and 18 UTC).

According to several authors, operational forecasts benefit from the combination of different models by considering different physical parameterisations, numerical schemes, model resolutions and forcings (Di Liberto et al., 2011; Rozante et al., 2014). In this study, dynamical downscaling is supplied by an ensemble of regional models implemented by the different research centres listed in Table 1. The reader is referred to the references listed in Table 1 for a more detailed description of the atmospheric models.

---

[1] Applied only at the 10 km resolution.
[2] Applied only at the 2 km resolution.



**Table 1.** List of the meteorological models implemented in this study.

| Meteorological model | Provider | Details | Reference |
|---|---|---|---|
| BOLAM[1] | CNR-ISAC (Italy) | Hydrostatic 60 vert. levels | Davolio et al. (2020) |
| MESO-NH | LAERO (France) | Non-hydrostatic 70 vert. levels | Lac et al. (2018) |
| Met Office Unified Model (MetUM) | UK Met Office | Non-hydrostatic 90 (70) vert. levels for 2 km (10 km) | Bush et al. (2020) for 2km Walters et al. (2017) for 10km |
| HARMONIE-AROME[2] | AEMET (Spain) | Non-hydrostat 65 vert. levels | Bengtsson et al. (2017) |
| WRF (5 variants) | CNR-ISAC1 (Italy) | Vers. 4.3 50 vert. levels | Miglietta et al. (2021) |
|  | CNR-ISAC2 (Italy) | Vers. 4.3 50 vert. levels | Avolio and Miglietta (2022) |
|  | AUTH (Greece) | Vers. 4.2.1 50 vert. levels | Carrió et al. (2019) |
|  | UIB (Spain) | Vers. 3.9.1 51 vert. levels | Pytharoulis et al. (2018) |
|  | UOA (Greece) | Vers. 4.1.3 55 vert. levels | https://forecast.uoa.gr |

The regional models have different characteristics, parametrizations and implementations but they share a common setup considering the following constraints:

80
- 5-day run covering the period from 15 (00 UTC) to 20 (00 UTC) September 2020;

- initial and boundary conditions from ECMWF IFS analysis;

- computational domain extending from approximately 28-43° N and 10-25° E;

- two horizontal grid spacing: 10 km (similar to the ECMWF IFS) and 2 km;

- deep convection is parameterized with 10 km and explicit with 2 km.

85    The objective of this study is not to mimic an operational forecasting system, but to analyse the level of uncertainty due to the meteorological mesoscale models, associated with this extreme event that is strongly sustained, especially in the initial phase, by deep convection. The common setup of the atmospheric models minimises the error in large-scale dynamics due to the boundary conditions (using the same analysis fields). Due to the common integration domain, the different results are mainly ascribable to the characteristic of the models. Moreover, to allow a fair comparison of the results at 10 and 2 km grid
90  spacing, the domain and initial/boundary conditions are also the same for all simulations.



## 2.2 Coupled hydrodynamic-wave model

The sea level and wave patterns during medicane Ianos were simulated using the finite element SHYFEM hydrodynamic model (https://github.com/SHYFEM-model/shyfem; Umgiesser et al., 2014) coupled with the unstructured WWMIII spectral wave model (Roland et al., 2009). The considered interactions between waves and surge were: (1) the contribution of waves to the total sea levels through the wave set-up and wave set-down; (2) the influence of tides and storm surge on wave propagation affecting the refraction, shoaling and breaking processes; (3) the effect of sea level variation and currents on the propagation, generation and decay of the wind waves. The coupled model has been already applied to simulate hydrodynamics and waves in the Mediterranean Sea (Ferrarin et al., 2013) and in the Adriatic Sea during extreme meteo-marine events (Roland et al., 2009; Cavaleri et al., 2019; Ferrarin et al., 2021).

Both the hydrodynamic and wave models run on the same computational domain that covers the whole Mediterranean Sea through an unstructured grid consisting of approximately 200,000 triangular elements (Fig. 1a). The use of elements of variable sizes is fully exploited to create a seamless transition between the open sea and the coastal waters following the medicane path. The ocean models solve the combined large-scale oceanic and small-scale coastal dynamics in the same discrete domain by subdivision of the basin into triangles varying in form and size. Mesh resolution varies from 12 km in the open sea to a couple of km along the cyclone path (Fig. 1b) and up to a few hundred metres along the western coast of Greece, where the cyclone landed (Fig. 1c). Model bathymetry over the Mediterranean Sea is obtained by a bilinear interpolation on the model grid of the European Marine Observation and Data Network (EMODnet) 2020 dataset (EMODnet Bathymetry Consortium, 2020).

The numerical experiments were forced by 10 m winds and mean sea level pressure (MSLP) provided by the ensemble of meteorological models described in section 2.1. ECMWF IFS 6-hour analyses were used to cover the wave-current model grid outside the atmosphere domain. Also, meteorological forcings from the ECMWF IFS analysis over the whole current-wave model domain are considered for benchmarking. All numerical experiments cover the period from 10 to 20 September 2020 with a spin-up period of 5 days (from 10 to 15 September 2020) for which we used ECMWF IFS 6-hour analysis as forcing. Since we are interested in examining only the meteorological contribution of the cyclone, we imposed no sea-level Atlantic Ocean boundary conditions at the Gibraltar Strait. The 2D barotropic hydrodynamic model simulates the sea level induced by the storm surge (driven by MSLP and wind) and waves, without considering tidal and baroclinic dynamics.

## 3 Results for Medicane Ianos

### 3.1 General meteorological conditions

Medicane Ianos formed in the central Mediterranean Sea over the warm waters of the Gulf of Sidra on 15 September 2020 developing from the interaction between an upper tropospheric precursor and a cluster of convective storms. Then the cyclone moved northward progressively deepening over the Ionian Sea. On 17 September 2020, Ianos evolved into a powerful medicane and on 18 September, it made landfall on the western coast of Greece (near Lefkada, Kefalonia and Zakynthos Islands), accompanied by high coastal waves and storm surges determining coastal damages and flooding (Zekkos et al., 2020; Lagou-



**Figure 1.** Bathymetry of the Mediterranean Sea with the unstructured grid superimposed: a) whole model domain, b) zoom over the Ionian Sea, c) zoom over the western Greek coast. The red point-line in panels a) and b) indicates the cyclone's track as defined by the 6-hour ECMWF IFS analysis from 2020-09-15 00 UTC to 2020-09-19 00 UTC. The red dots and the green triangle in panel c) mark the location of the sea level and wave monitoring stations, respectively.

vardos et al., 2022; Zimbo et al., 2022). The cyclone's path, as defined by the 6-hour ECMWF IFS analysis, is indicated with a red point-line in Fig. 1.





Since this study aims at evaluating the sea conditions driven by medicane Ianos, we focused the meteorological analysis only on the mean sea level pressure and wind over the sea, leaving out the physical processes behind the genesis and evolution of the medicane, as well as its effects overland (e.g. rainfall and damaging winds). Surface winds derived from the Advanced SCATterometer (ASCAT) onboard Metop satellites provided by the Royal Netherlands Meteorological Institute present an overview of the cyclone structure over the Ionian Sea (Fig. 2). On 16 September, the cyclone was located over the south-

western Ionian Sea with winds above 20 m s$^{-1}$ surrounding the centre. On 17 September, the cyclone intensified and in the evening it approached the western Greek coast, thus travelling over a distance of about 300 km in 23 hours at an average speed of 3.6 m s$^{-1}$. At this stage, southerly winds up to 25 m s$^{-1}$ hit the Zakynthos and Kefalonia islands. Ianos then remained over the Ionian islands during the night between 17 and 18 September maintaining high wind intensity.

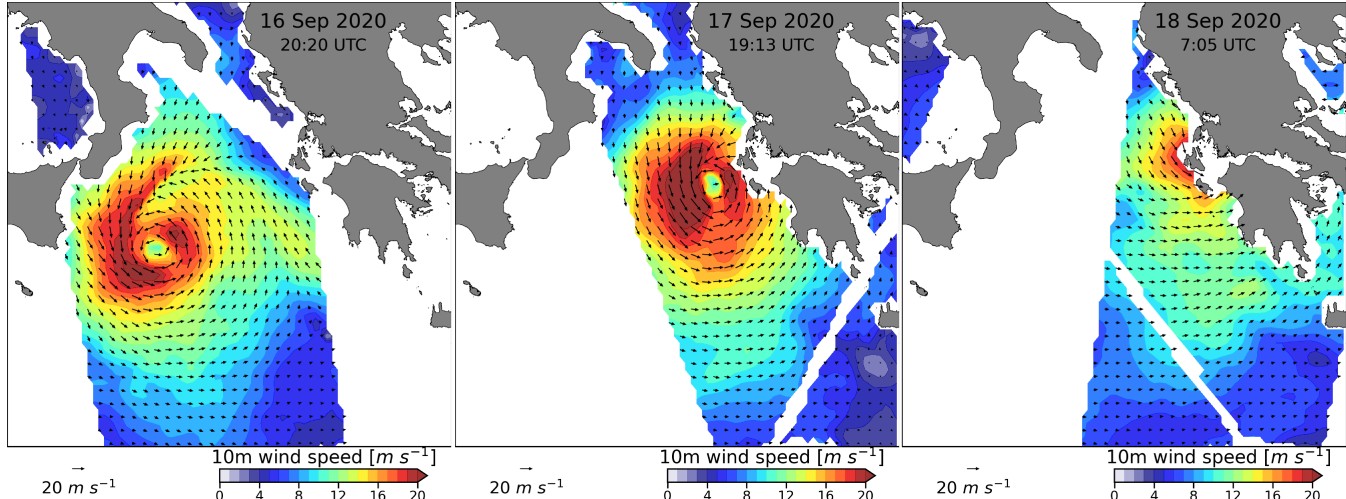

**Figure 2.** Scatterometer surface wind intensity (colour shaded) and direction (black arrows) fields on 16, 17 and 18 September 2020.

The results of the different atmosphere models are presented in Fig. 3 in terms of the cyclone's path and along-path minimum

pressure. Generally, the 2 km models produced tracks closer to analysis and a deeper cyclone with respect to the 10 km ones. While the location of landfall over Greece is shifted southeastward by all models compared to the analysis, both error and spread in tracks are about twice larger in 10 km than in 2 km simulations. Furthermore, a large spread is found in the simulated cyclone intensity and is also larger in 10 km than in 2 km simulations. The high resolution runs systematically overestimate the maximum intensity compared to the IFS analysis, although the latter underestimates the actual cyclone intensity (MSLP values

of 984.3 and 989.1 hPa were recorded in the landfall region; Lagouvardos et al., 2022). These results stress the high sensitivity of medicane Ianos to the resolved physical processes. Such uncertainty in the resolved track and intensity of the cyclone leads naturally to diverse estimations of ocean impacts which are assessed in the following sections.





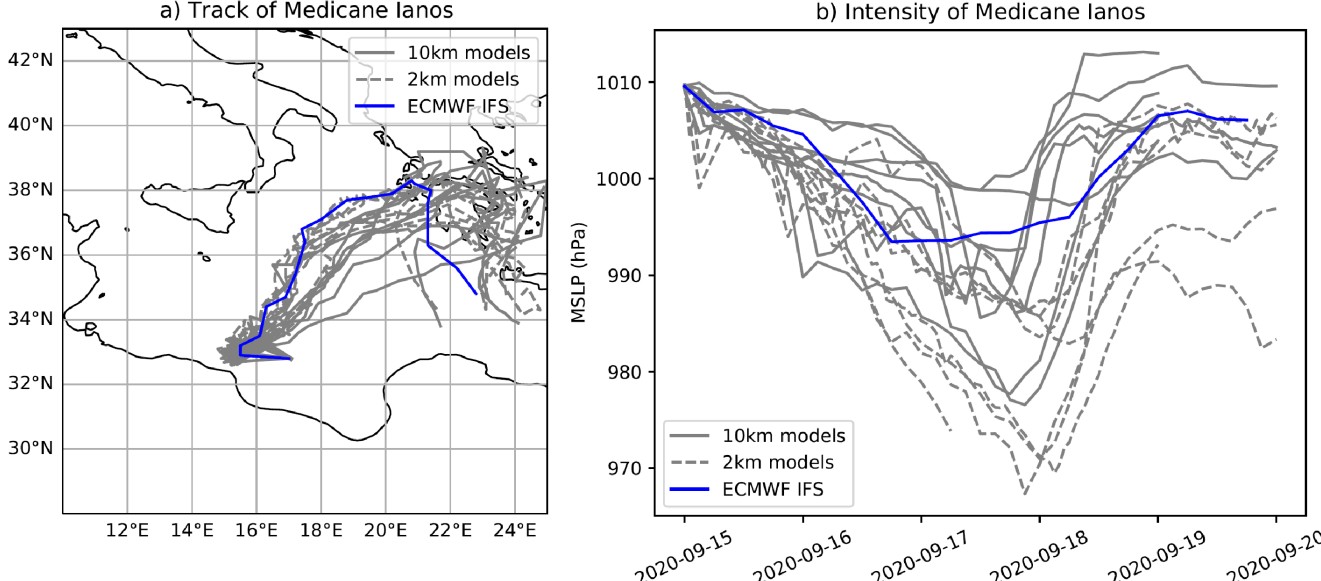

**Figure 3.** a) track (computed from the position of the minimum mean sea level pressure) and b) intensity (in terms of mean sea level pressure) of medicane Ianos as simulated by the different meteorological models.

## 3.2 Open sea conditions

Clearly, the strong surface winds swirling around the cyclone centre produced a severe sea state with high waves. Indeed, the
wave model forced by ECMWF IFS analysis simulated significant wave heights (SWH) with values up to 5 m in the Ionian
Sea (Fig. 4) and patterns replicating the surface wind fields detected by the scatterometers (see Fig. 2). It is worth noting that
waves generated by medicane Ianos propagated over the whole Ionian Sea determining maximum SWH values higher than 1.5
m along the southern Italian, the northern African and the western Greek coasts.

Satellite-based SWH observations confirmed the simulated values in the Ionian Sea. We considered SWH data acquired
by altimeters (Jason-3, Sentinel-3A, Sentinel-3B, Cryosat-2, SARAL/AltiKa, CFOSAT and Hai Yang-2B) passing over the
central Mediterranean Sea from 16 to 18 September 2020 (left panel in Fig. 5) and retrieved from the E.U. Copernicus Marine
Service (https://doi.org/10.48670/moi-00178). Along-track (at a spatial resolution of 7 km) satellite SWH observations were
used to assess the capacity of the wave model in reproducing the sea state during the medicane. In this study, we consider
normalized standard deviation, centered RMSE (root mean square error of the simulated values with respect to the observations
after subtracting the respective mean fields), bias (difference between the mean of simulation results and observations) and R
(correlation coefficient between model results and observations) as the metrics for measuring the model performance.

The statistical values shown in the Taylor diagram (Taylor, 2001, right panel of Fig. 5) and summarised in Table 2, indicate
that the wave simulations forced with the 10 km meteorological fields have a slightly lower RMSE with respect to the ones that
use the 2 km forcings (on average 1.0 m for 10 km and 1.1 m for 2 km). On the contrary, R is higher for the 2 km simulations





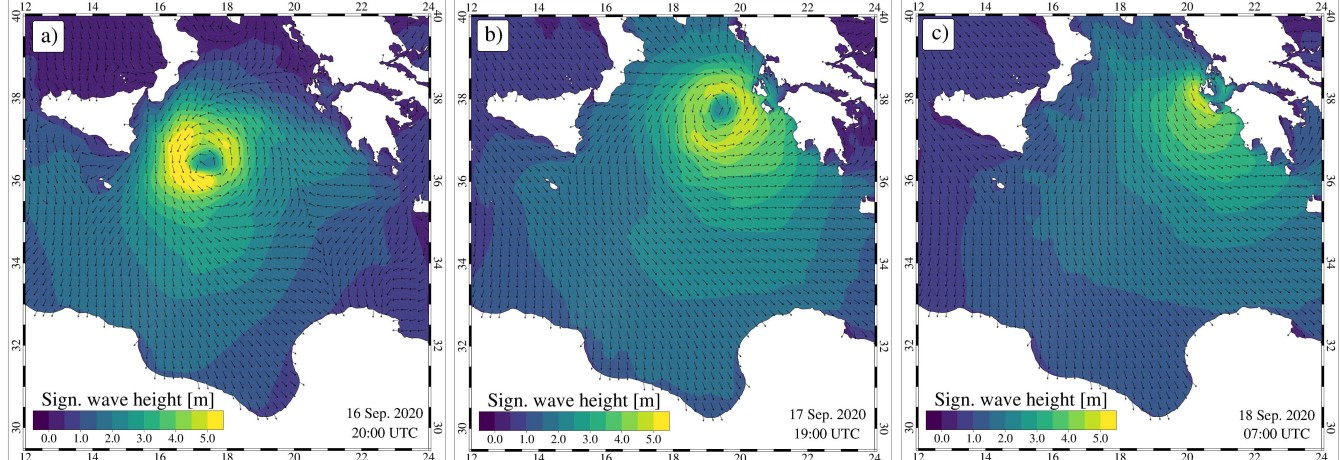

**Figure 4.** Significant wave height (colour shaded) and mean wave direction (black arrows) fields on 16, 17 and 18 September 2020 as simulated by the wave model forced with ECMWF IFS analysis.

**Table 2.** Statistical analysis (in terms of centered RMSE, BIAS and R) of simulated significant wave height in the open sea for different meteorological forcing. Statistics are reported for both the 10 km and 2 km meteorological forcing.

| Meteorological forcing | Centered RMSE (m) | BIAS (m) | R |
|---|---|---|---|
| | 10km / 2km | 10km / 2km | 10km / 2km |
| ECMWF IFS an. | 0.41 | 0.27 | 0.92 |
| BOLAM | 0.91 / – | -0.17 / – | 0.72 / – |
| MESO-NH | 0.69 / 1.03 | 0.34 / 0.66 | 0.90 / 0.88 |
| MetUM | 0.67 / 0.91 | -0.38 / 0.44 | 0.80 / 0.81 |
| HARMONIE-AROME | – / 1.56 | – / 1.07 | – / 0.83 |
| WRF-ISAC1 | 1.00 / 1.29 | 0.21 / 0.52 | 0.75 / 0.83 |
| WRF-ISAC2 | 0.66 / 0.94 | 0.05 / 0.40 | 0.82 / 0.80 |
| WRF-AUTH | 1.20 / 0.59 | -0.62 / -0.15 | 0.52 / 0.84 |
| WRF-UIB | 1.96 / 2.23 | 1.21 / 1.71 | 0.75 / 0.85 |
| WRF-UOA | 1.09 / 0.57 | 0.15 / 0.23 | 0.67 / 0.90 |

(on average 0.84 for 2 km and 0.74 for 10 km). Generally, both the 10 and 2 km sets of simulations have a standard deviation larger than observed and the 2 km numerical experiments tend to overestimate the measured SWH (the average BIAS for the 2 km runs amounts to 0.6 m). It is worth noting that none of the 10 and 2 km forcing simulations performs better than the one driven with the 6-hour ECMWF IFS analysis, which has an RMSE of 0.4 m and R equal to 0.92. This emphasises the challenge of accurately simulating the medicane evolution several days ahead, even on a limited area forced by the analysis at its lateral

boundaries.

The simulated sea levels forced by ECMWF IFS analysis show a circular pattern on 16 and 17 September following the medicane path with peak values in the centre of the cyclone of about 20 cm (Fig. 6). Such a distribution is mostly induced by





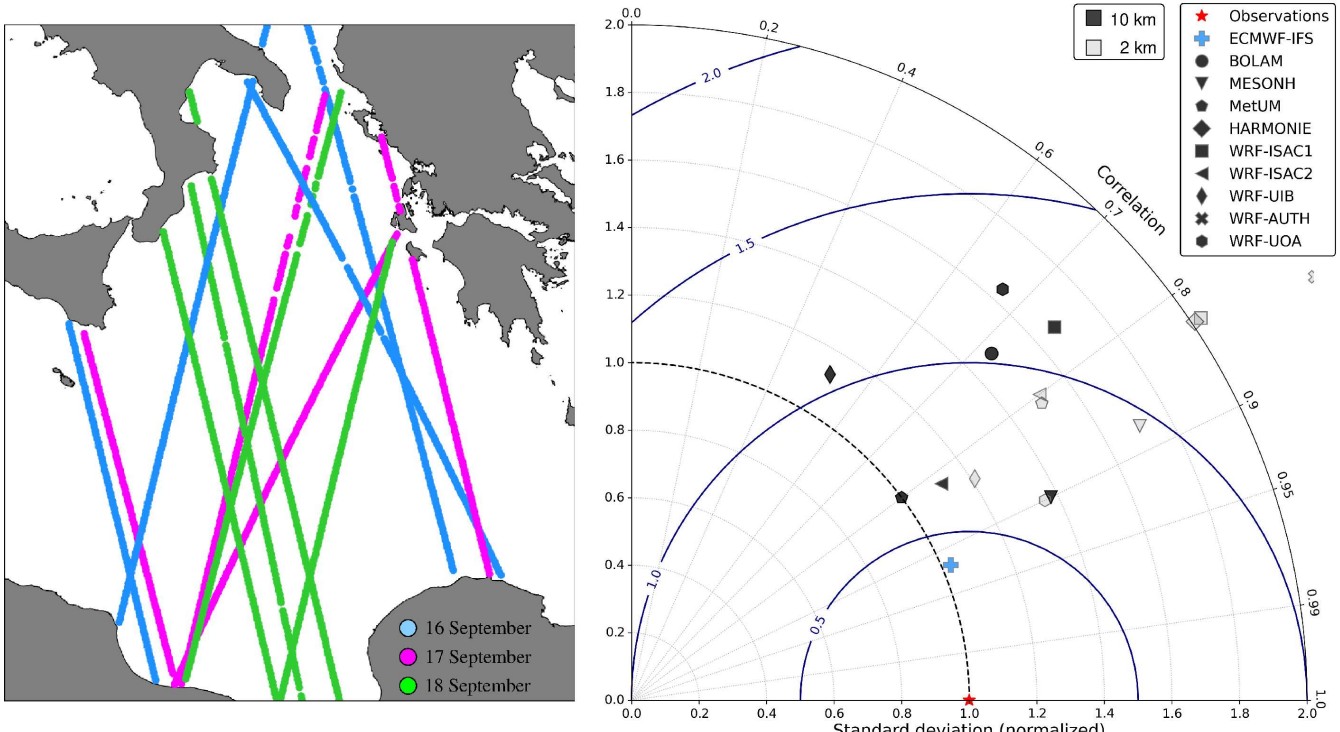

**Figure 5.** Left: tracks of satellites used for the open sea wave comparison. Right: Taylor diagram displaying a statistical comparison with satellite observations of simulated significant wave heights (filled and empty markers indicate 10 km and 2 km forcing, respectively). Some data points may lie outside the Taylor diagram.

the contribution associated with the sea level isostatic adjustment to the air pressure variations (the so-called inverse barometric effect which amounts to 1 cm of sea level per 1 hPa of air pressure). Wind and waves raised the sea levels along the northern

African coast with values up to 10 cm. Between 17 and 18 September, the cyclone hit the western Greek coast raising by more than 15 cm the sea levels near Lefkada, Zakynthos and Kefalonia islands.

The 1Hz (7 km) interval along-track sea surface heights recorded by altimeter Sentinel-3B (passage of 2020-09-17 20:00 UTC; magenta line in Fig. 6b) and Sentinel-3a (passage of 2020-09-18 09:00 UTC; green line in Fig. 6c), retrieved from the E.U. Copernicus Marine Service (https://doi.org/10.48670/moi-00140), were used for visual model comparison. We considered

the filtered sea surface height data uncorrected for the dynamic atmospheric component. It is important to stress that altimeters register the total sea surface height that is the result of different phenomena, including also the baroclinic dynamics which are not accounted for in the storm surge model (Cavaleri et al., 2021). Therefore, the along-track observations presented in Fig. 7 have higher spatial variability with respect to the model results. However, the rise in sea level towards the Ionian islands (in the order of 20 cm on 17 September and 10 cm on 18 September) is captured by both the storm surge model and the altimeters. As

for the waves, the model results display a large scatter of the simulated sea levels.





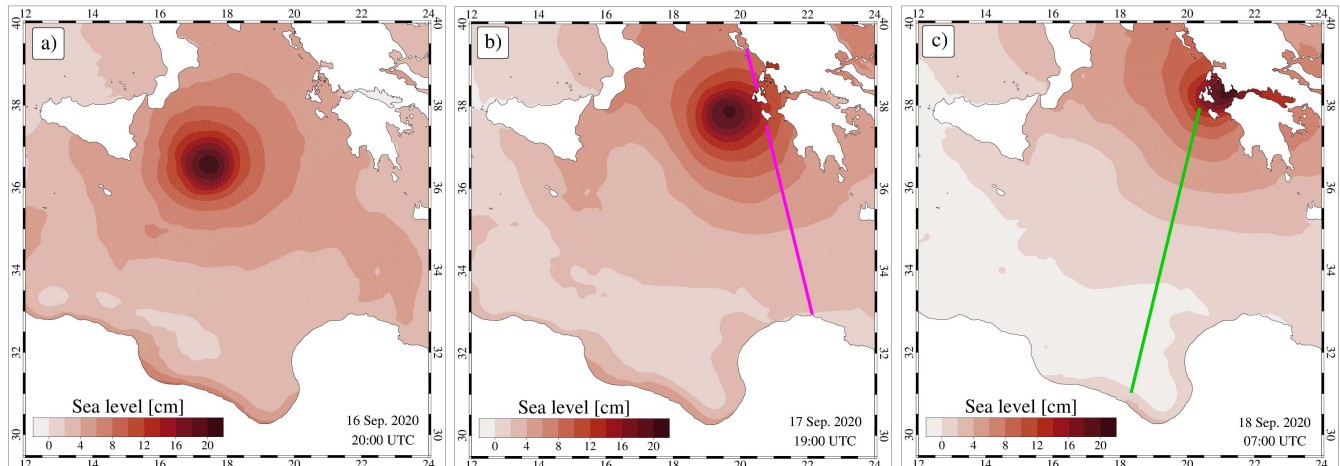

**Figure 6.** Sea level fields on 16, 17 and 18 September 2020 as simulated by the hydrodynamic model forced with ECMWF IFS wind and MSLP. The magenta and green lines indicate the altimeter tracks used for the model comparison presented in Fig. 7.

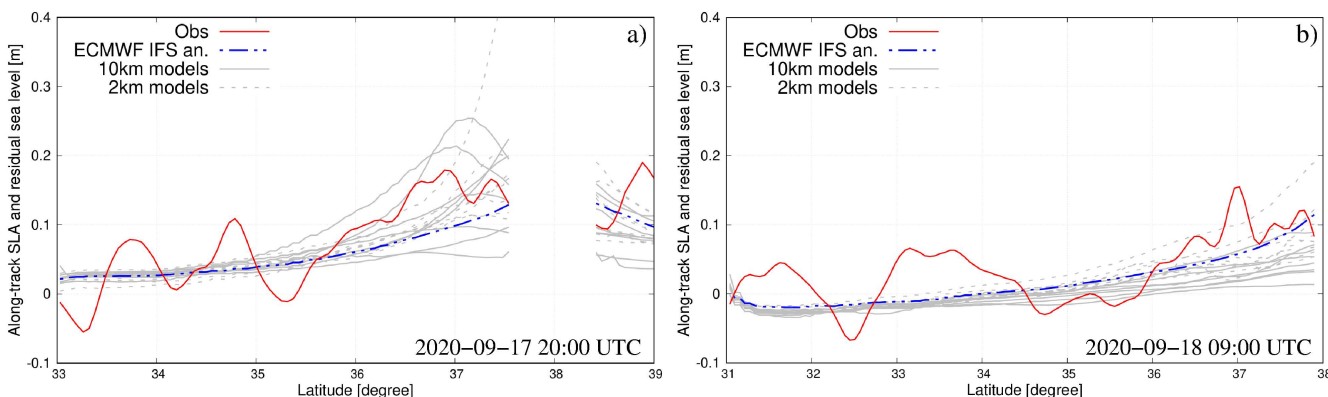

**Figure 7.** Along-track sea level anomalies (SLA) from altimeters and simulated sea levels. a) Sentinel-3B (2020-09-17 20:00 UTC; magenta line in Fig. 6b) and Sentinel-3a (2020-09-18 09:00 UTC; green line in Fig. 6c).

## 3.3 Sea conditions at the coast

While approaching the coast, the strong winds associated with medicane Ianos raised the sea levels near the Ionian Islands with the consequent flooding of several coastal locations (Zekkos et al., 2020). The sea level data acquired at the tide gauges of Zakynthos, Katakolo and Kyparissia (indicated with red dots in Fig. 1) and provided by the UNESCO/IOC sea level station monitoring service (http://www.ioc-sealevelmonitoring.org/), were processed with a tidal harmonic analysis tool based on the least squares fitting (Codiga, 2011) to separate the tidal and the atmospheric (storm surge) contributions to the total sea level. Observations clearly indicate a significant storm surge contribution associated with the medicane on 17 and 18 September with

peak values of 0.19, 0.27 and 0.16 m at Zakynthos, Katakolo and Kyparissia, respectively (Fig. 8a, b and c). The only available wave monitoring station in the area of interest is the buoy of Pylos (provided by the Poseidon monitoring, forecasting and

information system for the Greek seas; https://poseidon.hcmr.gr/) which recorded a maximum significant wave height of 4.7 m on 18 September at 03 UTC but probably missed the storm's peak (Fig. 8d).

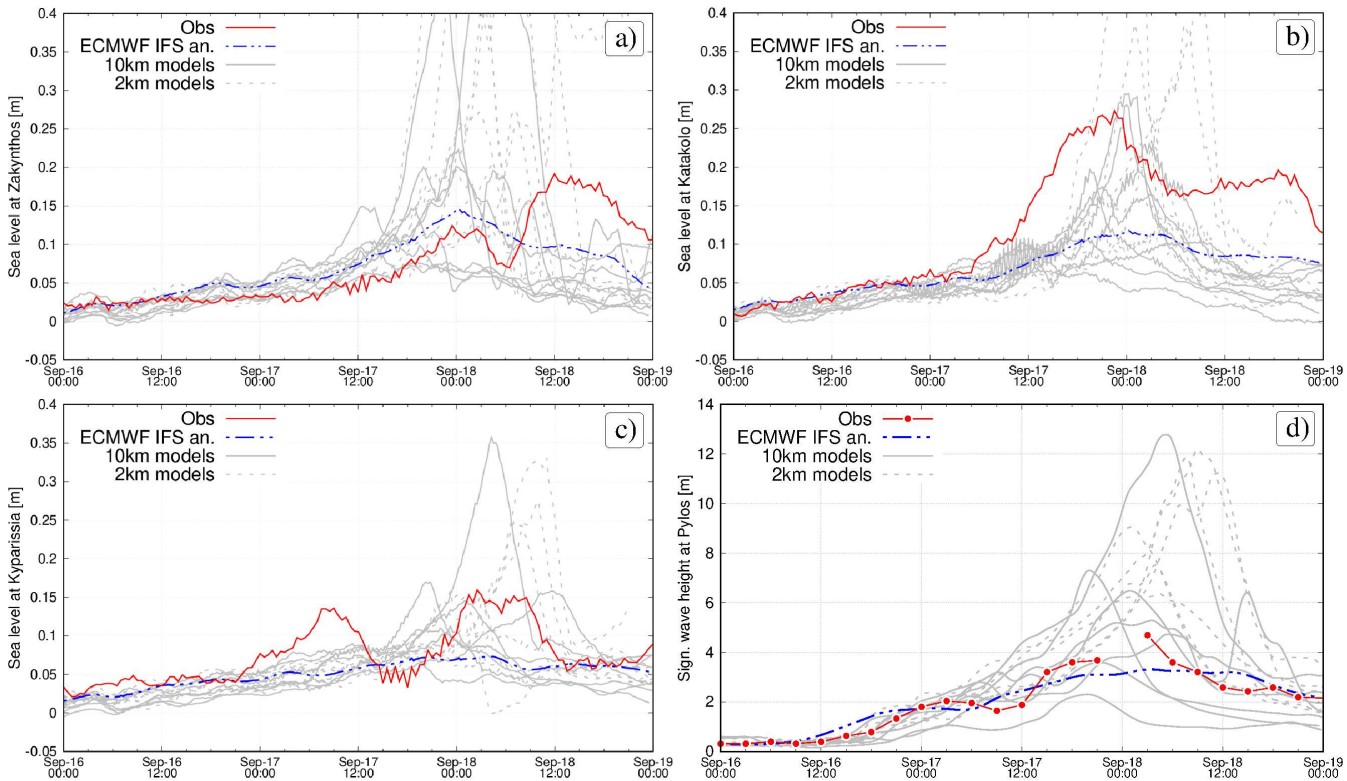

**Figure 8.** Observed and simulated sea levels at Zakynthos (a), Katakolo (b), Kyparissia (c) and significant wave height at Pylos (d).

The analysis of the numerical results highlights that even if the simulation forced by ECMWF IFS analysis correctly reproduced the wave conditions in the open sea, it strongly underestimated the peak values of both the sea level and the wave height at the coast. The storm surge from both the 2 and the 10 km forcing experiments show a large spread at the coast with some

simulations strongly overestimating the peak values. Similarly, the simulated significant wave height reaches maximum values ranging from 2 to more than 12 m at Pylos. Such a large spread in model results is mostly related to the complexity of the coastline. Indeed, the response of the sea to an intense cyclone with a highly variable wind pattern in space and time is strongly dependent on the specific meteorological and morphological conditions of the considered coastal location. Therefore, even a small deviation in the simulated cyclone path or intensity could greatly change the impact on the sea conditions. However,

given the long duration (5 days) of the simulations, it is not surprising to see such a large spread. Indeed, the results presented





**Table 3.** Observed and simulated peak values and timing (expressed as the ensemble mean and standard deviation) for the 10 and 2 km sets of simulations at the selected monitoring stations.

| Variable and station | Observations | | 10 km ensemble | | | | 2 km ensemble | | | |
|---|---|---|---|---|---|---|---|---|---|---|
| | Peak value | Peak time | Mean peak value | Peak spread value | Mean peak time | Peak spread time | Mean peak value | Peak spread value | Mean peak time | Peak spread time |
| Sea level at Zakynthos | 0.19 m | 18 Sep 15:00 | 0.16 m | 0.14 m | 18 Sep 03:45 | 6.7 h | 0.30 m | 0.20 m | 18 Sep 02:15 | 5.2 h |
| Sea level at Katakolo | 0.27 m | 17 Sep 22:30 | 0.24 m | 0.22 m | 18 Sep 01:00 | 5.8 h | 0.47 m | 0.34 m | 18 Sep 01:00 | 4.0 h |
| Sea level at Kyparissia | 0.16 m | 18 Sep 02:30 | 0.13 m | 0.10 m | 18 Sep 04:15 | 5.8 h | 0.15 m | 0.11 m | 18 Sep 08:30 | 4.3 h |
| Sign. wave height at Pylos | 4.7 m | 18 Sep 03:00 | 5.0 m | 3.6 m | 18 Sep 00:15 | 6.4 h | 8.1 m | 3.5 m | 18 Sep 06:45 | 3.4 h |

in Fig. 3 show high variability of the medicane track and intensity in the different meteorological models, that consequently simulated different locations (spanning over about 250 km of coastline) and timing of cyclones' landing.

In order to assess the uncertainty associated with the two sets of simulations (10 and 2 km grid resolution), we computed their ensemble mean and standard deviation (considered here as a measure of the ensemble spread) and we report in Table 3 the computed peak values and timing. We have to point out that the maximum value of the ensemble spread does not always coincide with the maximum value of the ensemble mean. These results highlight that the 2 km experiments generally produce a larger ensemble mean and spread with respect to the 10 km ones, but there is no clear indication that one of the simulation sets provides a more accurate reproduction of the observed peak sea levels and waves. The simulated maximum values of the ensemble mean are found to occur shifted by 2 to 10 hours with respect to the observed peaks with a spread in the timing ranging from 3.4 to 6.7 hours, depending on the location. It results that the 2 km experiments have a lower spread of the peak time (4.2 hours as the average of the selected stations) with respect to the 10 km ones (6.2 hours), thus reducing the uncertainty on the time occurrence of the storm peak.

## 4   Assessing the potential coastal hazard

Most of the uncertainty associated with the simulation of a sea storm event, like medicane Ianos, resides in the atmospheric forcing (Chaumillon et al., 2017; Cavaleri et al., 2018). We have seen in the previous sections that the different numerical experiments display a large spread of the hazardous sea storm conditions at the coast, for both the peak value and its timing. Indeed, a small deviation in the meteorological simulation (cyclone's trajectory and intensity) may produce a relevant effect on the sea level and wave reproduction, especially in areas characterised by a complex coastline. Moreover, the 2 km spatial high-resolution atmospheric forcings did not show a clear improvement in these aspects. However, the probabilistic approach





developed through the application of an ensemble of numerical experiments could provide insights into model uncertainty
when resolving storms and into the limits of the probabilistic forecast of impacts.

It is clear that a substantial warning of hazardous sea conditions could be available three days before the event if the oc-
currence of severe conditions was simulated by all models, as in the downscaling exercise discussed here. However, along a
complex coastline like the one of western Greece, a clear picture of the areas affected by severe waves and coastal flooding
cannot be provided with accuracy by a single numerical simulation. On the contrary, the ensemble predictions could be used
to assess the potential impact of the extreme event and eventually to allow an efficient flood risk management plan.

Following Ferrarin et al. (2020), here we used the ensemble mean and spread for defining sea level and wave hazard sce-
narios. In this study, we used all available numerical experiments (both the 10 and 2 km sets) for creating an ensemble of
16 members. To provide an overview of the expected potential hazard and its uncertainty associated with Ianos, the temporal
maximum values of the sea level and the SWH computed using the ensemble mean and spread are shown in Fig. 9 and Fig. 10,
respectively.

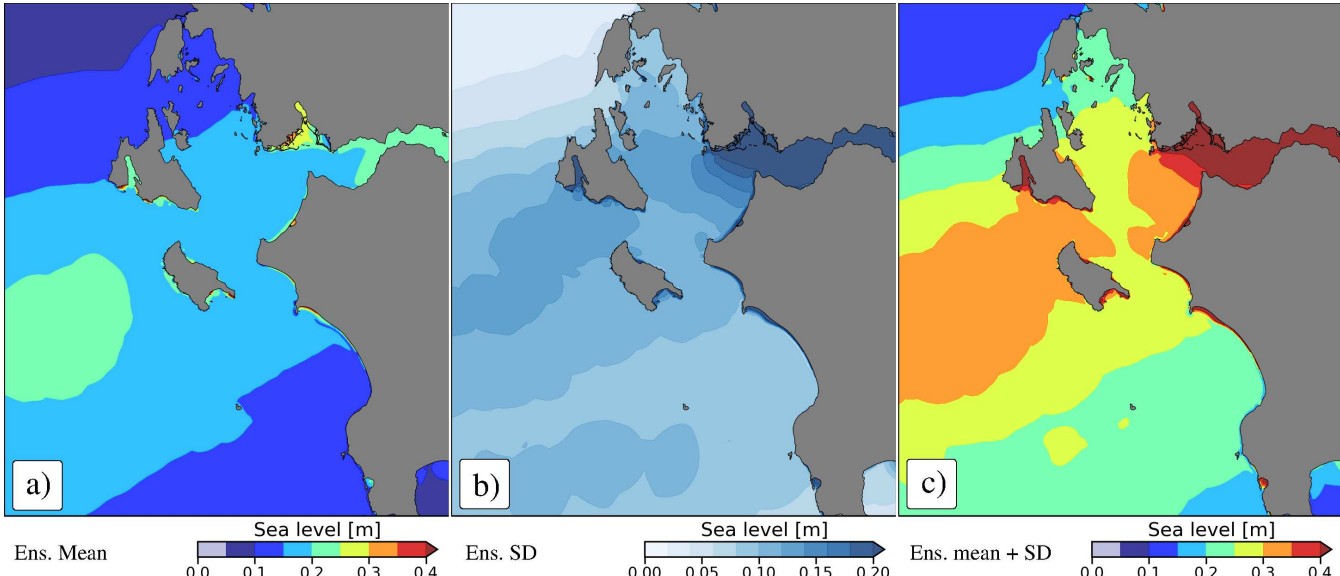

**Figure 9.** Maximum values of the storm surge ensemble mean (a), standard deviation (b) and the hazard scenario represented by the sum of
the ensemble mean and standard deviation (c).

The maximum storm surge values are found at specific coastal locations, mostly represented by small bays and headlands
where the combined action of wind and waves push the water to the coast (Fig. 9a). Similarly, the high ensemble standard
deviation is obtained for narrow bands at specific coastal segments (Fig. 9b). The ensemble mean may be considered as the
most probable hazard scenario to be used for determining the coastal areas potentially flooded during the event. However, due
to the large uncertainty in simulating such an intense event, a more conservative and safe hazard scenario could be provided by
the sum of the ensemble mean and the ensemble standard deviation (Fig. 9c).





Similar considerations can be derived for the SWH presented in Fig. 10. Medicane and other small-scale cyclones are potentially damaging events on coastal areas but the risk they impose could not be forecasted exactly (Ferrarin et al., 2020). The ensemble approach provides an overview of the coastal areas and infrastructures that can potentially be damaged by high waves, like the western side of Zakynthos island where waves may reach values from 5 m (Fig. 10a) in the mean scenario to 9 m (Fig. 10c) in the conservative scenario.

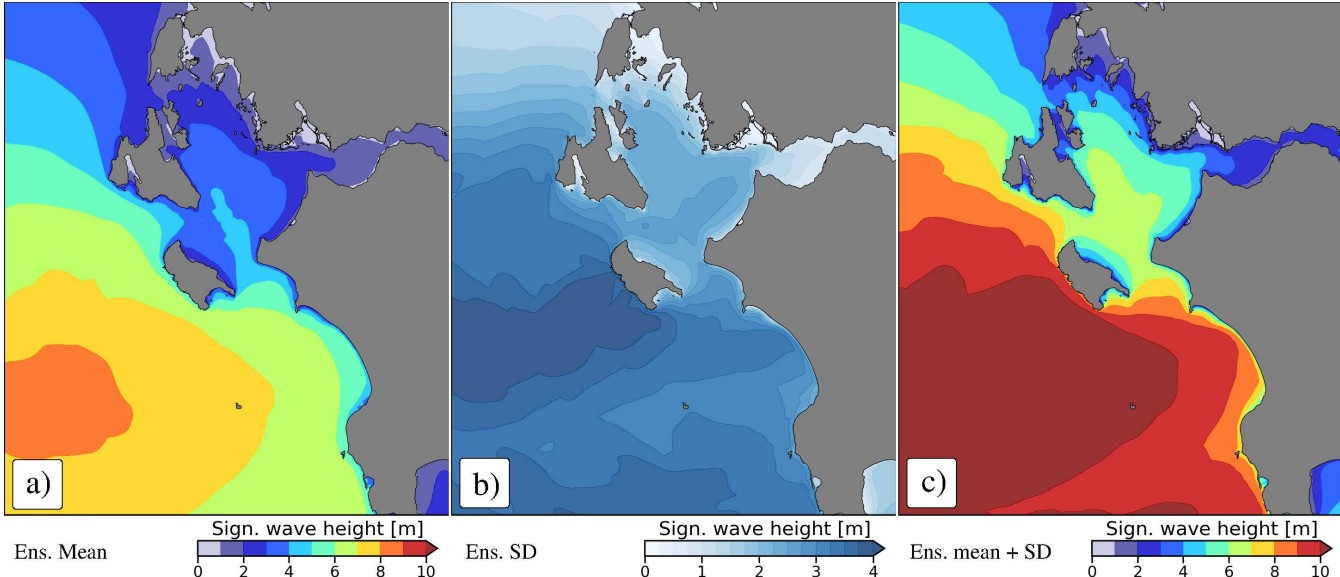

**Figure 10.** Maximum values of the sign. wave height ensemble mean (a), standard deviation (b) and the hazard scenario represented by the sum of the ensemble mean and standard deviation.

It is worth noting that every coastal location has site-specific flooding and damaging thresholds which depend on the coastal morphology and storm defences. Therefore, the hazard scenarios of sea levels and waves could be extracted for particular coastal locations and used in combination with threshold levels. For issuing flood risk bulletins, the nearshore total sea level should be considered, which at a given coastal location is the sum of several contributions, such as mean sea level variability, astronomical tide, storm surge and wave setup and run-up, acting on different temporal and spatial scales (Barnard et al., 2019; Woodworth et al., 2019; Ferrarin et al., 2022). Therefore, the simulated storm surge (which also includes the wave setup) has to be added to the site-specific astronomical tide and mean sea level to estimate the total sea level. Consequently, in addition to the uncertainty mentioned above on the cyclone's intensity and path, its time of landfall may also have a significant role in the impacts depending on the superposition with the tidal peak. In order to evaluate the total sea level at the monitoring stations, the ensemble storm surge results should be added to the periodic tidal contribution that can be predicted with great precision for months, using a good harmonic analysis tool. For example, we report in Fig. 11 the computed total sea levels at the Katakolo tide gauge station.



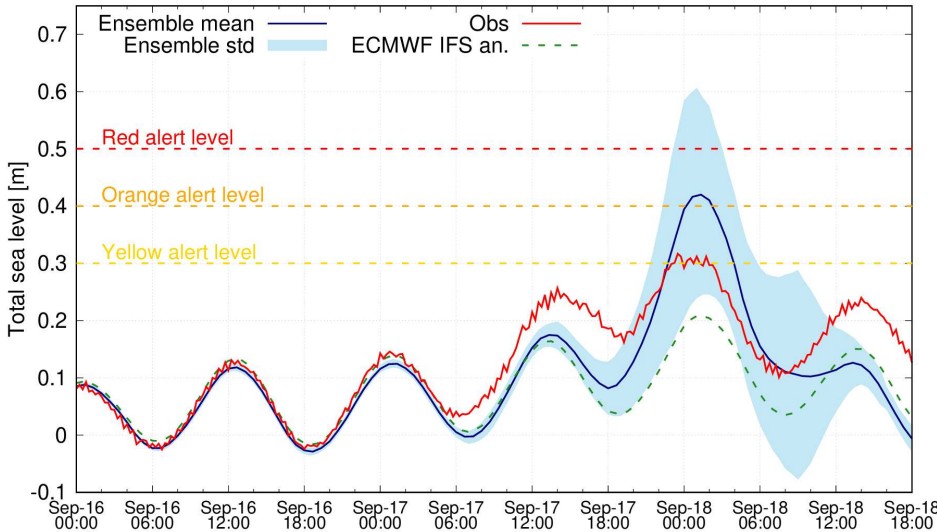

**Figure 11.** Observed and simulated total sea levels at Katakolo. The yellow, orange and red dashed lines indicate hypothetical thresholds for issuing flooding risk alerts.

Alerts usually refer to threshold levels associated with a site-specific low, mid and high risk of flooding. In the example reported in Fig. 11, we considered hypothetical flooding alert levels at 0.3, 0.4 and 0.5 m. In this case, the results forced by ECMWF IFS analysis underestimate the sea level at the selected location and do not indicate the exceedance of even the yellow flooding alert level. Conversely, the mean of the ensemble, even if overestimates the maximum sea level registered, indicates a high level of coastal risk. These results highlight the high level of uncertainty in simulating the total sea level at the coast,

with the most severe scenario (obtained by summing the ensemble mean and the ensemble spread) providing evidence of the occurrence of exceptional sea conditions.

## 5    Conclusions

In this study, the combined use of meteorological and ocean models enabled the analysis of extreme sea conditions driven by medicane Ianos, which hit the western coast of Greece on 18 September 2020, flooding and damaging several coastal locations.

The performed numerical experiments allowed the investigation of the propagation of uncertainty from the atmosphere to the ocean. The large spread associated with the meteorological ensemble highlighted the large model uncertainty in simulating such an extreme weather event. Even modest deviations in the cyclone's path and intensity may result in large changes in the coastal sea levels and wave height.

    The uncertainty in the amplitude, location and timing of the storm peak was analysed. Increasing the meteorological model

grid spacing from 10 to 2 km does not determine a clear improvement in the reproduction of the cyclone's impact on the sea conditions. Both the 10 and 2 km experiments show a large spread of the cyclone track and intensity, with the 2 km simulations



generally having deeper pressure minimums and stronger winds. This results in an overestimation of both the wave height and the sea levels in the Ionian Sea and along the western coast of Greece when using the 2 km meteorological fields as forcing for the ocean models. However, the 2 km experiments have a lower spread of the peak time with respect to the 10 km ones, thus reducing the uncertainty on the time occurrence of the storm peak.

Given the large model uncertainty associated with the reproduction of extreme meteo-marine events, we believe that the ensemble approach may provide very useful information on flood risk management plans. The different simulations have been combined to extract the ensemble mean and standard deviation (spread) used for outlining hazard scenarios. These simulated hazard conditions represent a fundamental component of the coastal risk assessment to be combined with the vulnerability and exposure of the specific coastal segment.

The meteorological simulations performed here are based on global ECMWF analysis and therefore do not represent the full predictability of medicane Ianos. They are closer to a downscaling of reanalysis, given the small integration domain. However, the multi-model / multi-physics approach can be easily extended to operational forecasting for providing in advance information on the coastal areas potentially affected by hazardous sea conditions. The ensemble results can be used to develop dynamic flood maps of specific coastal areas (Barnard et al., 2019), thus forecasting the potential flood impact of such extreme phenomena.

*Data availability.* This study has been conducted using E.U. Copernicus Marine Service Information: https://doi.org/10.48670/moi-00178, https://doi.org/10.48670/moi-00140. Tide gauge data of Zakynthos, Katakolo and Kyparissia are provided by the UNESCO/IOC sea level station monitoring service (http://www.ioc-sealevelmonitoring.org/index.php). The wave data acquired by the Pylos buoy are provided by the Poseidon monitoring, forecasting and information system for the Greek seas (https://poseidon.hcmr.gr/).

*Author contributions.* CF conceived the idea of the study with the support of FP, SD and EF. FP, SD, MMM, EA, DSC, IP, CS, PP and JJGA performed the meteorological simulations. CF and MB performed the hydrodynamic and wave numerical simulations. CF collected the observations, analyzed the sea level and wave results and prepared the figures (except Fig. 3 which has been produced by FP). All authors discussed, reviewed and edited the different versions of the manuscript.

*Competing interests.* The authors declare that they have no conflict of interest.

*Acknowledgements.* The authors wish to thank Dr Antonio Ricchi for providing the storage infrastructure and assistance. This work has been supported by the COST action CA19109 MedCyclones (European network for Mediterranean cyclones in weather and climate) and the Interreg Italy-Croatia STREAM project (Strategic development of flood management, project ID 10249186).



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
