# Peer review of "Assessing the coastal hazard of medicane Ianos through ensemble modelling"

_EGUsphere, 2022_

## Referee Comment (RC1)

Review of

Title: Assessing the coastal hazard of medicane Ianos through ensemble modelling
Author(s): Christian Ferrarin, Florian Pantillon, Silvio Davolio, Marco Bajo,
Mario Marcello Miglietta, Elenio Avolio, Diego S. Carrió, Ioannis Pytharoulis,
Claudio Sanchez, Platon Patlakas, Juan Jesús González-Alemán, and Emmanouil
Flaounas
MS No.: egusphere-2022-990
MS type: Research article
Iteration: Initial submission

The paper presents results of an ensemble of barotropic ocean models, coupled to a wave model and a number of atmospheric models, to give a probabilistic forecast of coastal sea levels in Greece during medicane Ianos.

The paper is clearly written and the Figures are clear. There are benefits (and pitfalls) of a large ensemble spread during an extreme event, but the most interesting takeaway to me seems to be that higher resolution is not automatically the holy grail of ocean modeling and will not, in itself, solve the prediction problem. I recommend publication after some minor revisions.

Specific remarks:

page10, L175: can the authors be more specific of what SLA product exactly they use. SLA netCDFs give sla_filtered variable, but according to the variable attribute, this is

sla_filtered:comment = "The sea level anomaly is the sea surface height above mean sea surface height; the uncorrected sla can be computed as follows: [uncorrected sla]=[sla from product]+[dac]+[ocean_tide]+[internal_tide]-[lwe]"

The authors state that they use sla_filtered, "uncorrected for the " dac. Can the authors please provide the exact arithmetic expression of what is included and what is subtracted in SLA that they use?

Page11, L185: perhaps it would be useful for readers to mention that this is a UTIDE package.

Figure 8: a-c: are red lines further filtered after subtraction of tides? I am surprised that UTIDE itself would make such a good detiding job…

page14, L237: yes, mean+stdev could provide a conservative estimate of risk – but they could also lead to so many false positives that the product would cease to be taken seriously by downstream stakeholders. Perhaps this could be mentioned as a downside of such conservative estimates. Figure would be a dramatic false positive if this conservative approach were to be used. There is an compromise to be found between model precision ()how many predicted floods occurred) and its recall (how many occurred floods were predicted). It is not obvious to me that simply adding mean+stdev leads to a good compromise.

Page15, L245: I suppose it would also be interesting to employ deep-learning classifiers instead of binary threshold based methods for this task...

---

## Author Comment (AC1)

**Responses to the Reviewer#1 Comments and Suggestions**

**Journal: Natural Hazards and Earth System Sciences (NHESS)**
**Manuscript number: egusphere-2022-990**
**Manuscript title: Assessing the coastal hazard of medicane Ianos through ensemble modelling**

The original Reviewer's comments and suggestions are shown in regular typeface, while our responses are shown in italics. The line and figures numbers we use refer to the revised document.

**R1.1** General comments: "The paper presents results of an ensemble of barotropic ocean models, coupled to a wave model and a number of atmospheric models, to give a probabilistic forecast of coastal sea levels in Greece during medicane Ianos.

The paper is clearly written and the Figures are clear. There are benefits (and pitfalls) of a large ensemble spread during an extreme event, but the most interesting takeaway to me seems to be that higher resolution is not automatically the holy grail of ocean modeling and will not, in itself, solve the prediction problem. I recommend publication after some minor revisions.

*Response: We appreciate the comments and we improved the manuscript following all reviewer's suggestions.*

**R1.2** page10, L175: can the authors be more specific of what SLA product exactly they use. SLA netCDFs give sla_filtered variable, but according to the variable attribute, this is

sla_filtered:comment = "The sea level anomaly is the sea surface height above mean sea surface height; the uncorrected sla can be computed as follows: [uncorrected sla]=[sla from product]+[dac] +[ocean_tide]+[internal_tide]-[lwe]"

The authors state that they use sla_filtered, "uncorrected for the " dac. Can the authors please provide the exact arithmetic expression of what is included and what is subtracted in SLA that they use?

*Response: We concur with the reviewer that the use of satellite sea-level products was poorly explained. In our study, we considered the filtered sea level anomaly (SLA) data uncorrected from the dynamic atmospheric component (i.e., $SLA_{unc}=SLA+DAC$). In the revised manuscript, we will improve the altimeter sea level data description and include the arithmetic expression used for computing the DAC uncorrected SLA.*

**R1.3** Page11, L185: perhaps it would be useful for readers to mention that this is a UTIDE package.

*Response: We will mention the UTIDE package.*

**R1.4** Figure 8: a-c: are red lines further filtered after subtraction of tides? I am surprised that UTIDE itself would make such a good detiding job ...

*Response: Yes, the red lines identify the residual sea levels obtained by detiding the sea level observations. We will improve the figure's caption to mention "residual sea levels" instead of just "sea levels".*

**R1.5** page14, L237: yes, mean+stdev could provide a conservative estimate of risk - but they could also lead to so many false positives that the product would cease to be taken seriously by downstream stakeholders. Perhaps this could be mentioned as a downside of such conservative estimates. Figure 11 would be a dramatic false positive if this conservative approach were to be used. There is an compromise to be found between model precision (how many predicted floods occurred) and its recall (how many occurred floods were predicted). It is not obvious to me that simply adding mean+stdev leads to a good compromise.

*Response: We thank the reviewer for highlighting this issue. The mean+stdev technique is just one of the approaches that can be derived from an ensemble of simulations. As mentioned by the referee, this approach can lead to many false positives. We will improve the discussion by mentioning the limits of such an approach. We will also state that more complex probabilistic approaches (e.g. probability of hazard threshold exceedance) should be further implemented knowing the site-specific characteristics of the different coastal segments.*

**R1.6** Page15, L245: I suppose it would also be interesting to employ deep-learning classifiers instead of binary threshold based methods for this task ...

*Response: We thank the reviewer for the suggestion. We will mention deep-learning classifiers as a methodology for flood hazard mapping and risk assessment. We kindly ask the referee to provide some references about deep-learning classifiers.*

---

## Author Response (AR1)

**Responses to the Referees' Comments and Suggestions**

**Journal: Natural Hazards and Earth System Sciences (NHESS)**
**Manuscript title: Assessing the coastal hazard of medicane Ianos through ensemble modelling**

The Authors thank the Reviewers for their valuable comments and suggestions: a point-to-point reply list is attached below. All the recommendations and advices are taken into due account, and possible critical issues are fixed. We believe that the (improved) revised manuscript might now be suitable for publication.

In particular, we improved the description of the satellite sea level observations used in this study and of the model ensemble characteristics and limits.

Below, the Reviewer's comments and suggestions are shown in regular typeface, while our responses are shown in italics. Also, the line and figure numbers refer to the revised document.

**Response to Referee#1**

**R1.1** General comments: "The paper presents results of an ensemble of barotropic ocean models, coupled to a wave model and a number of atmospheric models, to give a probabilistic forecast of coastal sea levels in Greece during medicane Ianos.

The paper is clearly written and the Figures are clear. There are benefits (and pitfalls) of a large ensemble spread during an extreme event, but the most interesting takeaway to me seems to be that higher resolution is not automatically the holy grail of ocean modeling and will not, in itself, solve the prediction problem. I recommend publication after some minor revisions.

*Response: We appreciate the comments and we improved the manuscript following all reviewer's suggestions.*

**R1.2** page10, L175: can the authors be more specific of what SLA product exactly they use. SLA netCDFs give sla_filtered variable, but according to the variable attribute, this is

sla_filtered:comment = "The sea level anomaly is the sea surface height above mean sea surface height; the uncorrected sla can be computed as follows: [uncorrected sla]=[sla from product]+[dac] +[ocean_tide]+[internal_tide]-[lwe]"

The authors state that they use sla_filtered, "uncorrected for the " dac. Can the authors please provide the exact arithmetic expression of what is included and what is subtracted in SLA that they use?

*Response: We concur with the reviewer that the use of satellite sea-level products was poorly explained. In our study, we considered the filtered sea level anomaly (SLA) data uncorrected from the dynamic atmospheric component (i.e., $SLA_{unc}=SLA+DAC$). In the revised manuscript, we improved the altimeter sea level data description and included the arithmetic expression used for computing the DAC uncorrected SLA (lines 177-180).*

**R1.3** Page11, L185: perhaps it would be useful for readers to mention that this is a UTIDE package.

*Response: We mentioned the UTIDE package at line 193.*

**R1.4** Figure 8: a-c: are red lines further filtered after subtraction of tides? I am surprised that UTIDE itself would make such a good detiding job ...

*Response: Yes, the red lines identify the residual sea levels obtained by detiding the sea level observations. We improved the figure's caption mentioning "residual sea levels" instead of just "sea levels".*

**R1.5** page14, L237: yes, mean+stdev could provide a conservative estimate of risk - but they could also lead to so many false positives that the product would cease to be taken seriously by downstream stakeholders. Perhaps this could be mentioned as a downside of such conservative estimates. Figure 11 would be a dramatic false positive if this conservative approach were to be used. There is an compromise to be found between model precision (how many predicted floods occurred) and its recall (how many occurred floods were predicted). It is not obvious to me that simply adding mean+stdev leads to a good compromise.

*Response: We thank the reviewer for highlighting this issue. The mean+stdev technique is just one of the approaches that can be derived from an ensemble of simulations. As mentioned by the referee, this approach can lead to many false positives. We improved the discussion by mentioning the limits of such an approach (lines 247-248) (see also the response to comment R3.9.4). Moreover, we added at lines 272-276 the following sentences mentioning other probabilistic approaches: "Other more complex probabilistic approaches (e.g. probability of hazard threshold exceedance; Biolchi et al., 2022) can be further implemented by having a larger number of members and knowing the site-specific characteristics of the different coastal segments. In this context, deep-learning classifiers instead of binary threshold-based methods can be implemented in dynamical flood hazard mapping and risk assessment (Bentivoglio et al., 2022, and references therein)."*

*This issue is also mentioned in the conclusions.*

**R1.6** Page15, L245: I suppose it would also be interesting to employ deep-learning classifiers instead of binary threshold based methods for this task ...

*Response: We thank the reviewer for the suggestion. See our response to comment R1.5.*

**Response to Referee#3**

**R3.GC** The authors compare the prediction of wave and sea level conditions during the impact of the Medicane Ianos simulated by a coupled hydrodynamic-wave model. The model is forced by a set of atmospheric conditions simulated by different atmospheric models with different spatial resolution. Authors propose to use the ensemble mean and deviation to predict induced hazards in the coast as a conservative way of approaching to hazard/risk mapping.

The paper is well written, it is well structured and results are clearly presented, with figures and tables being all of them relevant.

Despite this, the manuscript presents some points that need to be addressed before being considered for publication. In what follows, some comments and suggestions are given.

*Response: We appreciate the comments and we are very grateful for the preciseness of the remarks. We improved the manuscript following all reviewer's suggestions.*

**R3.1** Lines 55-59. Which is the objective of the paper? Is the objective to present/propose a methodological approach? Is to identify which model performs better? To identify the best resolution? To quantify model uncertainty? To avoid confusion, authors can include a sentence such as "The main aim of this work is ...".

*Response: We concur with the reviewer that the main aim of this study was not clearly stated in the introduction. We rewrote part of the introduction as (lines 52-57) "The main aim of this work is to investigate how the model uncertainty associated with the reproduction of such a severe event propagates from the atmosphere to the marine coastal areas. We used an ensemble of coupled ocean-wave simulations forced by atmospheric fields from a suite of numerical weather models. The ensemble approach allows the assessment of the potential coastal hazard associated with a medicane identifying the coastal areas most impacted by severe waves and storm surges. The developed methodology can be directly implemented in an early warning procedure for providing an estimation of the peak sea storm conditions to be used in coastal risk management."*

**R3.2** Line 64. Which was the criteria to select the used atmospheric models?

*Response: All the mesoscale models available within the research initiative fostered by the MedCyclones COST Action were used. They include some of the most widespread atmospheric mesoscale models used in academic and operational applications. Such a variety of independent models is suitable to explore the uncertainty associated with model errors (Bowler et al., 2008). These sentences have been added at lines 73-76.*

**R3.3** Line 67. Which was the criteria to select these two resolutions?

*Response: 10 km and 2 km are about the most common horizontal resolutions adopted in operational centres for numerical prediction over large domains and over regional areas, employing respectively parameterized or explicitly resolved convection. Intermediate horizontal resolutions between 10 and 2 km belong to the so-called gray zone, where deep convection is neither fully parameterized nor resolved, and should thus be avoided. These details have been added in the revised manuscript at lines 76-79.*

**R3.4** Line 89. Is the objective to make a fair comparison or, to look for the best results? Thus, would it not be better to use the best set of initial/boundary conditions for different model resolutions.

*Response: As specified above (response to comment R3.1) and as written at line XX, the main scope is to investigate the propagation of the model uncertainty from the atmosphere to the ocean in the prediction chain. Indeed, the set of initial/boundary conditions has been chosen to obtain a balance between good performance and sufficient spread in the ensemble. Therefore, we tried to disentangle as much as possible that portion of the uncertainty due to the model formulation (whether it is a different parameterization scheme in the same model (WRF) or a completely independent model). Along this line, exploiting IFS analysis as boundary conditions and adopting a relatively small integration domain reduce the uncertainty associated with large scale forcing. Of course, this setup does not allow to explore the uncertainty stemming from initial conditions, but this aspect is out of the scope of the present study (in fact we say at line XX that we do not mimic an operational forecasting system).*

**R3.5** Lines 92-95. Why did you use this set of models? Will the results be similar for a different set or will they depend on the models used? If so, a warning about this should be included in the discussion of results and conclusions.

*Response: We selected the coupled SHYFEM-WWMIII system because unstructured grid models allow resolving the combined large-scale oceanic and small-scale coastal dynamics in the same discrete domain (section 2.2). All simulation results are model-dependent. However, as stated in section 4, most of the uncertainty associated with the simulation of a sea storm event, like medicane Ianos, resides in the atmospheric forcing. Therefore, we could assume that using a different ocean-wave model system will produce similar results if driven by the same forcing and boundary conditions. We included the above statement at lines 222-224.*

**R3.6.1 General meteorological conditions:** Would it be relevant to cite in this section results obtained by Comellas Prat et al. (2021) doi.org/10.3390/rs13244984?

*Response: Thank you for the reference. Comellas Prat et al. (2021) paper explores the sensitivity of WRF simulations to microphysics and to different IC/BCs (ECMWF vs GFS). The sensitivity to IC/BCs dominates. This is in agreement with our statement at lines 143-144 "These results stress the high sensitivity of medicane Ianos to the resolved physical processes", since the differences between IFS and GFS driving fields are at relatively large scales, given the nature of the two global models.*

*We added at the end of the mentioned sentence "… as also pointed out by Comellas Prat et al. (2021)."*

**R3.6.2 General meteorological conditions:** Fig 3. Are you using ECMWF IFS as the measured conditions during the Medicane propagation? Is this referring to ERA-5 reanalysis? If yes, please mention it for readers not familiar with it.

*Response: We are using here IFS operational analysis as reference. Given the much lower resolution, ERA5 reanalysis further underestimates the cyclone intensity. Both are based on same model from the ECMWF but in different versions and resolutions. Since there are no observations available along the cyclone track, the analyses are affected by a relevant uncertainty in terms of cyclone intensity, as can be inferred by the comparison with available station along the coast of Greece. Moreover, differences among analysis issued by different centres (ECMWF, GFS, UKMO) can be relevant. Since IFS analysis was used to initialize the models, it was also used as reference.*

**R3.6.3 General meteorological conditions:** Lines 125-127. Even if your objective was not to simulate atmospheric conditions, a large deviation from the simulated conditions will have a significant impact on marine variables.

*Response: Our objective was to simulate and analyze the marine conditions during medicane Ianos. To attain this aim, we need of course to simulate the whole atmospheric conditions. Here we just say that we focus our attention on the MSLP and wind, which are the two variables most impacting the sea state. We are aware that differences in MSLP and winds are connected to differences also in the other atmospheric fields, but a deeper analysis of the reasons leading to differences in the simulations is out of the scope of the paper.*

*The impact of differences in the simulated cyclone track and intensity on marine variables is mentioned at lines 144-146 and deeply discussed in sections 3.2, 3.3 and 4.*

**R3.7.1 Open sea conditions**: When did you compare measured and modelled wave fields? At what time of the cyclone path? Is there any difference throughout the duration of the event?

*Response: As mentioned in section 3.2, the modelled wave fields were compared with Satellite-based SWH observations passing over the central Mediterranean Sea from 16 to 18 September 2020, therefore when the cyclone was moving from the central Ionian Sea to the Greek coast. We included some more details in the text at lines 153-156. Due to the limited number of satellite tracks available for this area in the selected period (Figure 5), we did not carry out separate analyses for different times of the cyclone evolution. However, the comparison with the Pylos wave buoy data (Figure 8d) allowed us to highlight the change of the model spread throughout the duration of the event.*

**R3.7.2 Open sea conditions**: Lines 176-178. This may imply that you are using an ocean model that cannot reproduce the effects captured by the only data source available to validate/compare model results. Do you have any idea of the potential magnitude of this effect under the conditions studied to compare with the measured sea level profile?

*Response: Yes, the adopted barotropic ocean model cannot reproduce the whole spectrum of processes influencing the sea level. However, as shown by Scicchitano et al. (2021), the storm surge induced by the inverse barometric effect and wind stress associated with a medicane represents the largest contribution to the temporary rise of the sea level during such events. Indeed, barotropic models have been widely used for simulating the sea level impact of different sea storms (Bajo et al., 2023, and references therein). These statements have been included in the revised manuscript.*

**R3.8.1 Sea conditions at the coast:** Line 211. "...reducing the uncertainty...". If we use the same criteria you use here to state that 2 km reduces the uncertainty, 10 km will do so for the sea level and wave maximum value. However, you mentioned earlier that there was no clear indication that one of the simulation sets would provide a more accurate reproduction of the observed maximum sea levels and waves.

*Response: We computed both the spread in the maximum values (Peak spread value in Table 3) and the spread in the time occurrence of the storm peak (Peak spread time in Table 3). In the mentioned sentence we are referring to this last quantity (the time occurrence of the storm peak), which is lower for the 2 km simulations. On the other side, as reported in the text, there is no clear indication that one of the simulation sets provides a more accurate reproduction of the maximum observed sea level and wave values. These sentences have been clarified (lines 216-220).*

**R3.9.1 Assessing the potential coastal hazard:** Line 219. "...probabilistic approach". Do you refer here to the number of models used to produce the ensemble? It would be enough to say the ensemble method. Probabilistic seems to suggest a large number of simulations which is not the case.

*Response: Following the reviewer's suggestion, we changed the text to "...ensemble method".*

**R3.9.2 Assessing the potential coastal hazard:** Line 221. See comments 2, 3, 4, 6.1 and 6.3. This can be the result of your previous choices.

*Response: Yes the ensemble results are somehow dependent on different choices, but these are not random (see responses to the comments above). We included a sentence in the Conclusions (lines 281-282) stating that the ensemble results are sensitive on the selection of models, resolutions and boundary conditions (see the response to comment R3.10.2).*

**R3.9.3 Assessing the potential coastal hazard:** Lines 234-235. The ensemble mean may be considered as the most probable hazard scenario in the case you are using proper models fed with proper initial/boundary conditions. If not, it is only representing the most probable scenario according to the used conditions and models (see comment 7.2).

*Response: The mentioned sentence have been modify as (lines 243-245) "Assuming that we used proper models fed with proper initial/boundary conditions, the ensemble mean may be considered as the most probable hazard scenario to be adopted for determining the coastal areas potentially flooded during the event."*

**R3.9.4 Assessing the potential coastal hazard:** Using the sum of the mean and deviation as the final hazard assessment is a conservative approach that is, of course, on the safe side. However, it can generate a number of false warnings that may affect the population's future response to real warnings.

*Response: We thank the reviewer for pointing out this important aspect. We included in section 4 (lines 247-248) the following sentence "It must be pointed out that such a conservative approach can generate a number of false warnings that may affect the population's future response to real warnings." See also the response to comment R1.5.*

**R3.10.1 Conclusions:** Adapt conclusions accordingly to any change resulting from the above comments.

*Response: The conclusions have been adapted considering the referees' comments. See also the responses to comments R1.5 and R3.10.2.*

**R3.10.2 Conclusions:** Lines 284-286. Are you referring to the proposed ensemble or are you talking in generic terms? I would say that you need to be very careful in the selection of models to be used, resolutions and boundary conditions.

*Response: We are here referring to the proposed ensemble. To stress the need of carefully selecting the models, resolutions and boundary conditions, we added in the Conclusions section (lines 281-282) the following sentence: "It must be pointed out that the ensemble results are sensitive to the construction of the ensemble members and therefore special attention must be taken in selecting the proper models, resolution and forcing, initial and boundary conditions".*

**References**

Bajo, M., Ferrarin, C., Umgiesser, G., Bonometto, A., and Coraci, E.: Modelling the barotropic sea level in the Mediterranean using data assimilation, accepted in Ocean Sci., https://doi.org/10.5194/egusphere-2022-1126, 2023.

Bentivoglio, R., Isufi, E., Jonkman, S. N., and Taormina, R.: Deep learning methods for flood mapping: a review of existing applications and future research directions, Hydrol. Earth Syst. Sci., 26, 4345–4378, https://doi.org/10.5194/hess-26-4345-2022, 2022.

Biolchi, L. G., Unguendoli, S., Bressan, L., Giambastiani, B. M. S., and Valentini, A.: Ensemble technique application to an XBeach-based coastal Early Warning System for the Northwest Adriatic Sea (Emilia-Romagna region, Italy), Coast. Eng, 173, 104 081, https://doi.org/10.1016/j.coastaleng.2022.104081, 2022.

Bowler, N. E., Arribas, A., and Mylne, K. R.: The benefits of multianalysis and poor man's ensembles, Mon. Weather Rev., 136, 4113 4 129, https://doi.org/10.1175/2008MWR2381.1, 2008.

Comellas Prat, A., Federico, S., Torcasio, R. C., D'Adderio, L. P., Dietrich, S., and Pane-grossi, G.: Evaluation of the Sensitivity of Medicane Ianos to Model Microphysics and Initial Conditions Using Satellite Measurements, Remote Sens., 13, https://doi.org/10.3390/rs13244984, 2021.

Scicchitano, G., Scardino, G., Monaco, C., Piscitelli, A., Milella, M., De Giosa, F., and Mastronuzzi, G.: Comparing impact effects of common storms and Medicanes along the coast of south-eastern Sicily, Mar. Geol., 439, 106 556, https://doi.org/10.1016/j.margeo.2021.106556, 2021.